# Motion Forecasting with Unlikelihood Training in Continuous Space

**Deyao Zhu**[1]     **Mohamed Zahran**[12]     **Li Erran Li**[3]*     **Mohamed Elhoseiny**[1]

[1] King Abdullah University of Science and Technology, [2] Udacity,
[3] AWS AI, Amazon and Columbia University

deyao.zhu@kaust.edu.sa     mohammed.zahran@udacity.com
erranlli@gmail.com     mohamed.elhoseiny@kaust.edu.sa

**Abstract:** Motion forecasting is essential for making safe and intelligent decisions in robotic applications such as autonomous driving. Existing methods often formulate it as a sequence-to-sequence prediction problem, solved in an encoder-decoder framework with a maximum likelihood estimation objective. State-of-the-art models leverage contextual information, including the map and states of surrounding agents. However, we observe that they still assign a high probability to unlikely trajectories resulting in unsafe behaviors, including road boundary violations. Orthogonally, we propose a new objective, unlikelihood training, which forces predicted trajectories that conflict with contextual information to be assigned a lower probability. We demonstrate that our method can improve state-of-art models' performance on the challenging nuScenes and Argoverse real-world trajectory forecasting datasets by avoiding up to 56% context-violated prediction and improving up to 9% prediction accuracy. Code is avaliable at
https://github.com/Vision-CAIR/UnlikelihoodMotionForecasting

**Keywords:** Motion Forecasting, Autonomous Driving

## 1 Introduction

For robotic applications deployed in the real world, the ability to foresee the future motions of agents in the surrounding environment plays an essential role in safe and intelligent decision-making. In the autonomous driving domain, to predict nearby agents' future trajectories accurately, an agent must consider contextual information such as their past trajectories, potential interactions, and map information. State of the art prediction models [1, 2, 3] directly take contextual information as part of their input and use techniques such as graph neural networks to extract high-level features for prediction. They are typically trained with a maximum likelihood estimation (MLE) objective of ground truth trajectories in the predicted distribution. Although MLE loss encourages the prediction to be geometrically close to the ground truth, there is no explicit motivation to learn a distribution that leverages contextual information. These models may predict trajectories that violate the contextual information (e.g., go to opposite driving direction or out of the driving area) but still closes to ground truth. In contrast, humans can quickly notice that these trajectories are unlikely in a specific context. This observation suggests that using MLE loss cannot fully leverage contextual information, and hence may predict unsafe trajectories more likely.

To address the problem, existing methods like [4, 5], using contextual information, either introduce new learning parameters or apply reinforcement learning methods requiring further reward engineering. We propose a novel and simple method, unlikelihood training, that injects contextual information as a learning signal. Our loss penalizes the trajectories that violate the contextual information, dubbed as negative trajectories, by minimizing their likelihood in the predicted distribution. To generate negative or unlikely trajectories, we first draw many candidate trajectories from our model's predicted distribution. Then, we introduce a context checker to identify the trajectories that violate contextual information as negative trajectories[2]. The model is then encouraged to use contextual information to

---

*Work done outside of Amazon.

[2]the context checker is not required to be differentiable

5th Conference on Robot Learning (CoRL 2021), London, UK.

avoid predictions that violate context by minimizing the likelihood of negative trajectories. Hence, the predicted trajectories can be more accurate and, more importantly, safer.

Unlikelihood training [6] has been applied to neural language generation in discrete space. *To the best of our knowledge, we are the first to propose unlikelihood training for continuous space of trajectories.* For the discrete space of token sequences, repeating tokens or n-grams in the generated sequence are chosen as negative tokens. In contrast, we design a context checker to select negative trajectories sampled from the continuous distribution of model predictions as input to an unlikelihood loss. In this case, our learning signals contain not only the information of the ground truth trajectories from the MLE, but also the the context information from our proposed loss. Our method can be viewed as a simple add-on to models that estimate the distribution of future trajectories. It improves their performance by encouraging models to focus more on contextual information without increasing the number of model parameters. Our contributions are summarized as follows:

- We propose a novel and simple method, continuous unlikelihood training for motion forecasting in autonomous driving that encourages models to use contextual information by minimizing the likelihood of context-violated trajectories. Our method can be incorporated into state-of-the-art models that predict the future as distributions.

- Our large-scale experimental results on two challenging real-world trajectory forecasting datasets, nuScenes, and Argoverse, show that continuous unlikelihood training can improve the quality of the predicted distribution by avoiding maximally 56% context-violated prediction and improving 9% prediction performance.

## 2 Related Work

**Trajectory Forecasting**   Trajectory forecasting of dynamic agents, a core problem for robotic applications such as autonomous driving and social robots, has been well studied in the literature. State-of-the-art models solve it as a sequence-to-sequence multi-modal prediction problem [7, 8, 9, 3, 10, 2, 11, 1, 12]. MTP [8] and MultiPath [9] predict multiple future trajectories without learning low dimensional latent agent behaviors. Recent works including [7, 10, 3, 12] encode agent behaviors in continuous low dimensional latent space while MFP [2] and Trajectron++ [1] use discrete latent variables. Discrete latent variables succinctly capture semantically meaningful modes such as turn left, turn right. MFP [2] and Trajectron++ [1] learn discrete latent variables without explicit labels. They adopt a maximum likelihood estimation (MLE) objective or its approximations (e.g., VAE). In this paper, we show that MLE loss can ignore contextual information such as maps and states of surrounding agents. As a result, models with MLE loss only may assign a very high probability to unlikely trajectories. To alleviate this issue, R2P2 [13] trains a surrogate ground truth distribution first and use it to regularize the training of the predictor. However, the surrogate is still trained by MLE loss only.  In contrast, we propose an unlikelihood training objective that is based directly on the context information to reduce unlikely prediction, and we show that models with the maximum likelihood estimation objective can benefit from it.

**Unlikelihood Training.** In the context of language generation, recent works including [6, 14] propose unlikelihood training as a new method to utilize negative language data, which includes common degenerate cases like token repetition and distribution mismatch. The distribution mismatches occur as an artifact of beam search, which is biased to high-frequency tokens compared to low-frequency tokens which appear rarely. The method minimizes the likelihood of negative tokens to improve text generation quality in addition to maximizing the likelihood of the ground truth tokens. However, their approach operates in the discrete space. In contrast, our proposed method works in the continuous space of trajectories. We model negative data by a context checker that we propose, which operates on predictions to populate unlikely trajectories. Predicted trajectories that violate the checker are selected to leverage contextual information by minimizing their likelihood objective. We show the generality of our approach on two state-of-the-art distribution-based approaches: Trajectron++ [1] and a distribution-based variant of LaneGCN [15].

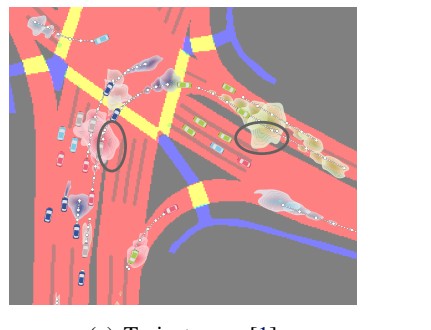 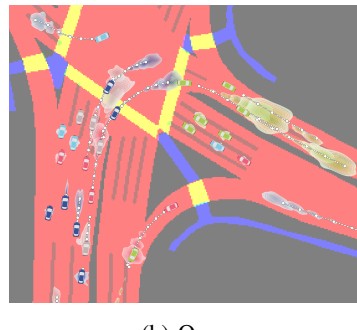

(a) Trajectron++[1]       (b) Ours

Figure 1: Examples of predicted distribution in a complex scenario from models trained with MLE only (a) and with our unlikelihood loss additionally (b). Some of the Trajectron++'s predicted distributions are out of the drivable region or cover the lane in the wrong direction (Highlighted by ellipses). Our method helps alleviate this issue. Predicted distribution is plotted as the colored region. White points denote the ground truth trajectories. More qualitative results are in appendix.

## 3 Method

### 3.1 Problem Definition

We aim at predicting the future trajectory $Y_i$ more accurately of a vehicle $i$ given input $X_i$. $X_i$ can include related information like rasterized maps or past positions of vehicle $i$ and surrounding agents. Because of the generality of our training mechanism, we skip the specific choice of this additional information and denote it as $X_i$ for conciseness. Due to different driving strategies, driving intents, and the complex traffic environment, there are usually multiple possible future trajectories given an input $X_i$; despite that there is only one ground truth future trajectory $Y_{i,gt}$ in a dataset recorded in the real world. To facilitate predicting multiple modalities, most state of the art methods [1, 2] model a distribution of possible future trajectories $p_\theta(Y_i \mid X_i)$ to cover all the possibilities given the input $X_i$ instead of predicting one trajectory. $\theta$ denotes the learning parameters of the model. Training such models can be done by Maximum Likelihood Estimation (MLE) that directly maximizes the likelihood of ground truth trajectory $Y_{i,gt}$ or its lower bound in the predicted distribution.

**Limitation of MLE on Motion Forecasting** MLE encourages the model to predict a distribution that allocates reasonable probability mass to the region where $Y_{i,gt}$ is located by minimizing the KL-divergence between the ground truth distribution and the predicted distribution. Because the trajectory distribution domain is over the geometric locations, MLE makes these two distributions "close" to each other geometrically. However, we argue that maintaining the geometrical nearness only is not good enough for motion forecasting tasks in autonomous driving. In complex traffic scenarios, there can be many potential trajectories close enough to the ground truth geometrically but are very unlikely to happen. For example, if the ground truth trajectory $Y_{i,gt}$ is on the outermost lane, a trajectory that is close to $Y_{i,gt}$ but outside the drivable region is unlikely to happen in the real world. MLE loss will not impose a significant enough penalty to avoid such a prediction. Fig.1(a) demonstrates a prediction example from an MLE-based method Trajectron++ [1]. Part of the distribution in this example is outside of the derivable region or on the lane in the wrong direction. The MLE-based loss only offers learning signals that contain the geometric location information of the ground truth trajectories. All the other contextual information like the drivable region is missing in this learning signal. Therefore, the model is not encouraged during training to utilize the rich contextual information in the input to avoid predictions that are geometrically close to ground truth but violates the context. In contrast, this is quite a simple task for humans.

### 3.2 Unlikelihood Loss

To mitigate this problem, we design a new loss term that encourages the model to consider contextual information more. Inspired by the positive and negative data pairs in contrastive learning and unlikelihood training, we additionally train our model to minimize the likelihood of trajectories that violate the contextual information given input $X_i$. We denote them as negative trajectories $Y_{i,neg}$.

Let's first assume that we already have a distribution of negative trajectories $p_{\mathrm{neg}}(\boldsymbol{Y}_i \mid \boldsymbol{X}_i)$. One intuitive way is to directly minimize the log likelihood of $\boldsymbol{Y}_{i,neg}$ in our predicted distribution, similar to MLE but in an opposite manner, as shown in Eq.1. We call it unlikelihood loss. We use a coefficient $\gamma$ to balance $L_{\mathrm{unlike}}$. Our loss can be combined with models that predict trajectory distribution as output. Eq.2 shows the final training objective.

$$L_{\mathrm{unlike}} = \mathbb{E}_{\boldsymbol{X}_i, \sim \mathbb{D}, \boldsymbol{Y}_{i,neg} \sim p_{\mathrm{neg}}(\boldsymbol{Y}_i|\boldsymbol{X}_i)}[\log p_\theta(\boldsymbol{Y}_{i,neg} \mid \boldsymbol{X}_i)] \tag{1}$$

$$L = L_{\mathrm{orig}} + \gamma L_{\mathrm{unlike}} \tag{2}$$

$L_{\mathrm{orig}}$ indicates the original training objective of the method we combine with. $L_{\mathrm{unlike}}$ helps leverage the context into the learning objective, force the model to better extract and use contextual information in $\boldsymbol{X}_i$, and generate a more reasonable distribution to avoid high $L_{\mathrm{unlike}}$.

### 3.3 Negative Trajectories

Our proposed loss term is highly dependent on the negative samples $\boldsymbol{Y}_{i,neg}$ from the distribution $p_{\mathrm{neg}}(\boldsymbol{Y}_i \mid \boldsymbol{X}_i)$. However, these are not given in the dataset. To solve this issue, we approximate the samples by directly drawing a set of trajectories from the predicted distribution and select the trajectories that violate the contextual information out by a context checker. Note that this checker does not need to be differentiable and it can be as complex and advanced as necessary. The type of unlikely predictions the model learns to avoid by our method depends on the type of unlikely trajectories the checker can identify.

**Context Checker Design.** We provide and implement a map-based checker design here to judge whether a given trajectory is compliant with context and road rules. In detail, the checker examines whether the trajectory goes into the lane in the opposite direction or out of the road. We create a map that stores the lane direction at every location of lanes and the drivable region. Two examples are shown in Fig.2. We first check whether all the locations of a given trajectory are in the drivable region. If so, we further calculate angles between velocity and the lane direction at each time step to see whether they are all within 90 degree. The velocities are approximated by differentiating the trajectory. The trajectories that fail to pass the test are identified as negative trajectories $\boldsymbol{Y}_{i,neg}$. Note that the lane direction information might be incomplete in some examples in the dataset, and in the case, we only use the drivable region information. In addition, human drivers also go against the context occasionally. We disable $L_{\mathrm{unlike}}$ if the ground truth trajectory violates the context to allow a similar prediction. Although this checker is not perfect, it is able to select out meaningful negative trajectories to support the training. Note that recent works like Beelines [16] provide other methods to detect the unlikely trajectories and it is possible to adopt them as additional rules in our checker. We leave it for future research.

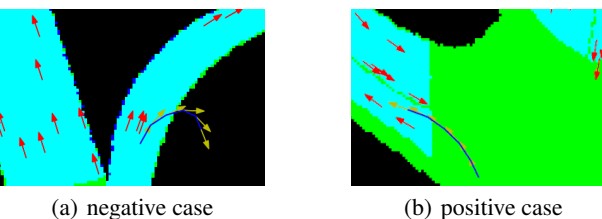

(a) negative case        (b) positive case

Figure 2: Examples of maps used in the checker in nuScences [17]. The green and blue regions together denote the drivable region. The lane direction is available in blue region. The lane directions are plotted as red arrows at random location. The blue line denotes the trajectory to check. Its velocity directions are represented as yellow arrows. (a) shows a negative trajectory that goes out of the road. (b) shows a positive trajectory that passes the checker.

### 3.4 Algorithm

The final algorithm is shown in Alg.1. At each iteration, we first run the forward pass of the model to get the output distribution $p_\theta(\boldsymbol{Y}_i \mid \boldsymbol{X}_i)$ given the input $\boldsymbol{X}_i$. Then, $K$ negative candidate trajectories are drawn from this distribution and we select the negative trajectories out $\boldsymbol{Y}_{i,neg}$ via our checker. After that, ground truth trajectory and negative trajectories are used to calculate the loss function, and

the model is updated. Finally, we proceed to the next iteration. Note that if there are no $\boldsymbol{Y}_{i,neg}$ in the $K$ negative candidates judged by our checker for data $i$, we don't apply $L_{\text{unlike}}$ on $i$.

## 3.5 Gradient Analysis

Assume a single-mode prediction case that the future position $\boldsymbol{y}_{i,gt,t}$ at time step $t$ of agent $i$ is modeled by a simple Gaussian distribution $\mathcal{N}(\boldsymbol{y}_{i,t}; \hat{\boldsymbol{\mu}}_{i,t}, \hat{\sigma}_{i,t}\boldsymbol{I})$. $\hat{\boldsymbol{\mu}}$ and $\hat{\sigma}_{i,t}$ are calculated by the model. With a single negative position $\boldsymbol{y}_{i,neg,t}$ we define a simple loss for step $t$ $L_t = -\log\mathcal{N}(\boldsymbol{y}_{gt,t}; \hat{\boldsymbol{\mu}}_t, \hat{\sigma}_t\boldsymbol{I}) + \log\mathcal{N}(\boldsymbol{y}_{neg,t}; \hat{\boldsymbol{\mu}}_t, \hat{\sigma}_t\boldsymbol{I})$ and omit the subscript $i$ for brevity. The gradient of $L_t$ with respect to $\hat{\boldsymbol{\mu}}_t$ and $\hat{\sigma}_t$ in this case is (derivation in supplementary):

$$\frac{\partial L_t}{\partial \hat{\boldsymbol{\mu}}_t} = -\frac{1}{\hat{\sigma}_t^2}((\boldsymbol{y}_{gt,t} - \hat{\boldsymbol{\mu}}_t) + (\hat{\boldsymbol{\mu}}_t - \boldsymbol{y}_{neg,t})) \tag{3}$$

$$\frac{\partial L}{\partial \hat{\sigma}_t} = -\frac{1}{\hat{\sigma}_t^3}(||\boldsymbol{y}_{gt,t} - \hat{\boldsymbol{\mu}}_t||^2 - ||\boldsymbol{y}_{neg,t} - \hat{\boldsymbol{\mu}}_t||^2) \tag{4}$$

Eq.3 shows that the center of the predicted distribution $\hat{\boldsymbol{\mu}}_t$ is pushed towards $\boldsymbol{y}_{gt,t}$ and pushed away from $\boldsymbol{y}_{neg,t}$ by this learning objective. In Eq.4, when $\boldsymbol{y}_{gt,t}$ is closer to the center than $\boldsymbol{y}_{neg,t}$, $\frac{\partial L}{\partial \hat{\sigma}_t}$ is positive and $\hat{\sigma}_t$ is decreased. Note that $\boldsymbol{y}_{neg,t}$ is selected out from samples of $\mathcal{N}(\hat{\boldsymbol{\mu}}_t, \hat{\sigma}_t\boldsymbol{I})$, this means when $\mathcal{N}(\hat{\boldsymbol{\mu}}_t, \hat{\sigma}_t\boldsymbol{I})$ covers context-violated region and this region is farther than ground truth region, $\mathcal{N}(\hat{\boldsymbol{\mu}}_t, \hat{\sigma}_t\boldsymbol{I})$ will shrink to exclude the negative region and become a better estimation to the true data distribution.

**Soft $\gamma$.** At the beginning of training, the model performs poorly and may generate bad predictions far away from the ground truth $\boldsymbol{y}_{gt,t}$. In this case, the standard deviation $\hat{\sigma}_t$ will increase according to Eq.4 and make the prediction more uncertain. To alleviate this issue, we turn on $L_{\text{unlike}}$ smoothly after the first few training epochs by making $\gamma$ in Eq.2 as a Sigmoid function of the training epoch centered at a specified epoch as a warm-up.

---

**Algorithm 1:** Training process

Initialize the model parameters $\theta$, learning rate $\alpha$ and coefficient $\gamma$
**while** *not converge* **do**
  $\mathbf{X}_i, \mathbf{Y}_{i,gt} \sim \mathbb{D}$
  Predict distribution $p_\theta(\mathbf{Y}_i \mid \mathbf{X}_i)$
  Draw K trajectories
   $\mathbf{Y}_{i,k} \sim p_\theta(\mathbf{Y}_i \mid \mathbf{X}_i)$
  Select $\mathbf{Y}_{i,neg}$ from $\mathbf{Y}_{i,k}$ via checker
  Compute original loss $L_{\text{orig}}$
  Compute $L_{\text{unlike}}$ using Eq.1
  $L = L_{\text{orig}} + \gamma L_{\text{unlike}}$
  $\theta = \theta - \alpha\nabla_\theta(L)$
**end**

---

## 4 Experimental Results

In this section, we present the experimental results of our method. Our method can be applied to the state of the art models that generate a future trajectory distribution and further improve the predicted distribution quality. We evaluated our method on nuScenes dataset [18] applied to Trajectron++ [1] and Argoverse dataset [19] applied to a distribution-based variant of LaneGCN [15].

**Evaluation Metrics** We use average $l_2$ displacement error (ADE) and final $l_2$ displacement error (FDE) to evaluate the prediction. We target improving the quality of the predicted distribution. To quantify the distribution quality, we randomly sample 200 trajectories from the predicted distribution and average ADE and FDE over these samples as the metric ADE-Full and FDE-Full. In addition, we report the context-violation rate of these 200 trajectories as measured by the context checker.

### 4.1 Experiments on nuScenes Dataset

nuScenes dataset [18] contains 1000 city driving scenes from both left-hand (Singapore) and right-hand (Boston) traffic regions. Each scene is 20s long and recorded at 2Hz. nuScenes is one of the most prominent open-source motion forecasting datasets with detailed semantic maps.

**Test Model** Trajectron++ [1] is a CVAE-based [20] model. Its input $\boldsymbol{X}_i$ contains positions, velocities, headings of the predicted and surrounding vehicles, and a map patch. The output distribution $p_\theta(\boldsymbol{Y}_i \mid \boldsymbol{X}_i) = \sum_{\boldsymbol{z}} p_{\theta 1}(\boldsymbol{z} \mid \boldsymbol{X}_i)p_{\theta 2}(\boldsymbol{Y}_i \mid \boldsymbol{X}_i, \boldsymbol{z})$ is a Gaussian mixture model with 25 components and modeled by an encoder net $p_{\theta 1}(\boldsymbol{z} \mid \boldsymbol{X}_i)$ and a decoder net $p_{\theta 1}(\boldsymbol{Y}_i \mid \boldsymbol{X}_i, \boldsymbol{z})$. In addition, it has

Table 1: **nuScenes:** Experimental results on nuScenes dataset [18] with Trajectron++ [1]. Our proposed loss avoids 16% context-violated prediction and improves the prediction accuracy performance of Trajectron++ by more than 8%, which indicates a better-predicted distribution. Mean and standard deviation are calculated over 5 runs.

| Model | FDE-Full | ADE-Full | Context-Violation-Rate | minFDE10 | minADE10 |
|---|---|---|---|---|---|
| Ground Truth | - | - | 5.44% | - | - |
| Trajectron++ | 2.74±0.10 | 1.04±0.05 | 10.59%±0.54% | 1.68±0.05 | 0.68±0.02 |
| **Trajectron++ with $L_{unlike}$** | **2.51±0.06** | **0.95±0.03** | **8.85% ± 0.32%** | **1.52±0.07** | **0.61±0.02** |
| Relative Improvement | +8% | +9% | + 16% | +10% | +10% |

Table 2: **nuScenes:** Experimental results on the nuScenes dataset for single prediction. FDE and ADE for all methods here are computed by only one predicted trajectory. For both Trajectron++ and our method, this trajectory is sampled by greedy search. Our method helps to improve the predicted accuracy. Mean and standard deviation are calculated over 5 runs.

| Model | FDE-1 | | | |
|---|---|---|---|---|
| | 1s | 2s | 3s | 4s |
| Const. Velocity [1] | 0.32 | 0.89 | 1.70 | 2.73 |
| S-LSTM [21] | 0.47 | - | 1.61 | - |
| CSP [22] | 0.46 | 2.35 | 1.50 | - |
| CAR-Net [23] | 0.38 | - | 1.35 | - |
| SpAGNN [24] | 0.36 | - | 1.23 | - |
| Trajectron++ [1] | 0.06±0.01 | 0.43±0.01 | 1.08±0.04 | 2.05±0.08 |
| **Trajectron++ with $L_{unlike}$** | **0.05±0.00** | **0.42±0.01** | **1.05±0.02** | **1.99±0.05** |

another encoder net $q_{\theta 3}(z \mid X_i, Y_{i,gt})$ used only in training. The original learning objective is shown in supplementary materials. $I_q$ denotes the mutual information.

**Experiments** The batch size is set to 1024. Models are trained for 35 epochs and we test the weights from the best epoch measured by average ADE on the validation set. The coefficient $\gamma$ in Alg.1 increases gradually from 0 to 1 as a sigmoid function centered at the 24th epoch. The initial learning rate is 3e-3 and it decays exponentially by 0.9995 per iteration. These hyperparameters except $\gamma$ are optimized for Trajectron++ and lead to better performance than that in the original paper. To increase the numerical stability of the training, we add a small constant $\epsilon = 1e - 9$ in Eq.1 to avoid the infinite gradient region of log function. In this case, Eq.1 becomes:

$$L_{\text{unlike}} = \mathbb{E}_{X_i, \sim \mathbb{D}, Y_{i,neg} \sim p_{\text{neg}}(Y_i|X)}[\log(p_\theta(Y_{i,neg} \mid X_i) + \epsilon))] \tag{5}$$

Mathematically, $\epsilon$ term scales the original gradient by $\frac{p_\theta(Y_{i,neg}|X_i)}{p_\theta(Y_{i,neg}|X_i)+\epsilon}$. Proof is shown in supplementary materials. In addition, we rotate the scenes randomly from $15°$ to $345°$ in the training set for data augmentation following the setting of the original Trajectron++. For each model, we run five experiments and report the mean and standard deviation of the measured metrics. The models are trained to predict 3 seconds into the future. To evaluate generalization beyond the training horizon, we test on both 3 second and 4-second prediction horizons.

Tab.1 shows the quantitative results of the 4s prediction measured by the FDE/ADE-Full metrics and the context-violation rate. Our unlikelihood loss encourages the prediction to respect context information more measured by the reduction of the context violation rate from 10.59% to 8.85%. Besides, our method increases the prediction accuracy as both FDE and ADE are reduced by about 8%. Results indicate that our method helps improve the quality of the predicted distribution from both context-compliant and accuracy perspectives. This is also demonstrated in the qualitative comparison in Fig.1. The predicted distribution from our methods covers the not-drivable and wrong direction region less. In contrast, original Trajectron++ tends to violate the contextual information when the prediction horizon is long. This shows that our method encourages the model to be more sensitive to the road boundary and the lane direction. Comparison with other methods is shown in Tab.2. The FDE for the single predicted trajectory is also improved by our method.

### 4.2   Experiments on Argoverse Dataset

Argoverse dataset [19] contains 300,000 5-second tracked scenarios in 2 American cities Miami and Pittsburgh with semantic maps. The data is recorded at 10 Hz. The first 2 seconds are used as input to predict the next 3 seconds' future.

Table 3: **Argoverse:** Experimental results on Argoverse dataset. Compared to our base model Gaussian LaneGCN, our proposed method reduces the context violation by a large margin and avoid 56% of prediction that violates context. ADE-Full and FDE-Full are reduced by 8% and 6%, respectively. Results indicate a better-predicted distribution with our unlikelihood loss $L_{unlike}$.

| Model | ADE-Full | FDE-Full | Context-Vio. | ADE-1 | FDE-1 |
|---|---|---|---|---|---|
| Ground Truth | - | - | 0.85% | - | - |
| Gaussian LaneGCN | 1.81 | 3.49 | 9.0% | 1.38 | 3.03 |
| **Gaussian LaneGCN with** $L_{unlike}$ | **1.67** | **3.29** | **4.0%** | 1.38 | **3.01** |
| Relative Improvement | +8% | +6% | + 56% | 0% | +1% |

**Test Model**   LaneGCN [15] is one of the state of the art models on Argoverse dataset. It takes the historical trajectories and a graph representation of lanes as input and directly generates 6 deterministic trajectories as the prediction of each vehicle. The predicted trajectories are trained by regression using smooth L1 loss. To investigate the effectiveness of our method on distribution-based models, we applied our method to a distribution-based variant of LaneGCN that predicts 6 trajectory distributions instead. The predicted position at each timestep is modeled by a simple isotropic Gaussian distribution with predicted variance. This is done by extending the dimension of the final output layer to include a log variance prediction additionally for each timestep. In addition, the model is modified to predict velocity distribution. The trajectory distribution is calculated by the integration over the velocity distribution. The smooth L1 regression loss is replaced by the minus log likelihood of the ground truth future (MLE training). The original training objective is shown in supplementary materials. This variant is denoted as Gaussian LaneGCN.

**Experiments**   Argoverse does not release the ground truth future trajectories for the test dataset. Therefore, we use the original validation set as our test set in this experiment. For both Gaussian LaneGCN and our method, we set the batch size to 128. The initial learning rate is 1e-3 and it decays to 1e-4 after 32 epochs. The model is trained for 36 epochs. The coefficient $\gamma$ of the unlikelihood loss in Alg.1 increases gradually from 0 to 1 as a sigmoid function centered at the 12th epoch. The results are listed in Tab.3. With our unlikelihood training loss, we significantly reduce the context violation rate from 9.0% to 4.0% compared to the original Gaussian LaneGCN. 56% of predictions that violate the context are avoided with our method. In addition, ADE-Full and FDE-Full are also reduced by 8% and 6%, respectively. Results indicate that our method successfully improves the predicted distribution quality by encouraging the model to focus on context information more.

### 4.3   Ablation Study

**The Influence of Our Loss Without Map Input**   Here we remove the map input from the model and demonstrate the influence of our unlikelihood loss in nuScenes in Tab.4. Interestingly, our loss helps improve the performance from 2.89 to 2.67 measured by the FDE-Full although the model does not see the context during inference. We think the reason is that compared to the Trajectron++ without map input, our unlikelihood loss still offers context information to support the training since this loss is calculated using the context. Therefore, our model receives more information during training and performs better. Besides, our method without map input even achieves a comparable result (FDE-Full 2.67) compared to Trajectron++ with map input (FDE-Full 2.74). This experiment shows that our loss can inject context information into learning signals. In addition, using map input only as the context information source improves FDE-Full of Trajectron++ by 0.15 (from 2.89 to 2.74). In contrast, if we use both map input and our unlikelihood loss , the performance is improved by 0.38 (from 2.89 to 2.51). Our loss doubles the contribution of context information.

**Remove Context-Violated Prediction Directly**   An intuitive way to avoid context-violated predictions is simply removing these predictions as a post-processing step. Even though this is straightforward, the prediction model in this case isn't trained to obtain the context violation knowledge. Our method encourages the model to understand the context better and to avoid the context violation by itself. Therefore, our model is more 'intelligent' compared to the method directly removing the context-violated prediction and leads to a better prediction. In addition, context violation is sometimes needed. A small part of the ground truth trajectories also violates the context. The model trained by our method can still give us context-violated prediction as our context violation rate is not zero in Tab.1. We list the ADE/FDE-Full of Trajectron++, Trajectron++ removing the context-violated prediction

Table 4: The effect of $L_{unlike}$ without map input to the model on nuScenes. Compared to Trajectron++ without map input, our loss performs better, which indicates that $L_{unlike}$ offers context information for the training.

| Map Input | $L_{unlike}$ | FDE-Full | ADE-Full |
|-----------|--------------|----------|----------|
| - | - | 2.89±0.11 | 1.08±0.03 |
| - | + | **2.67±0.08** | **1.01±0.02** |
| + | - | 2.74±0.10 | 1.04±0.05 |
| + | + | **2.51±0.06** | **0.95±0.03** |

Table 5: The performance when simply removing the context-violated prediction. Although this straightforward method improves the original baseline, our method achieves the best scores. This indicates that our unlikelihood loss helps models to understand the context better.

| Method | ADE-Full | FDE-Full |
|--------|----------|----------|
| Trajectron++ | $1.04 \pm 0.05$ | $2.74 \pm 0.10$ |
| Trajectron++ with Removing Violation | $0.98 \pm 0.04$ | $2.55 \pm 0.10$ |
| **Trajectron++ with $L_{unlike}$** | **$0.95 \pm 0.03$** | **$2.51 \pm 0.06$** |

Table 6: Ablation study on extended check-horizon of negative candidates on nuScenes dataset. When the horizon is properly extended from 3s to 4s, the checker can recognize more negative trajectories and improves the prediction accuracy. Evaluation horizon is 4s. Results are averaged over 2 runs.

| Model | Check-Horizon | ADE-Full | FDE-Full | Cotext-Vio. | ADE-1 | FDE-1 |
|-------|---------------|----------|----------|-------------|-------|-------|
| Trajectron++ | - | 1.04 | 2.74 | 10.6% | 0.77 | 2.05 |
| Trajectron++ with $L_{unlike}$ | 3s | 0.95 | 2.51 | 8.9% | **0.74** | 1.99 |
| Trajectron++ with $L_{unlike}$ | 4s | **0.92** | **2.43** | **8.5%** | 0.75 | **1.98** |
| Trajectron++ with $L_{unlike}$ | 5s | 0.96 | 2.58 | 9.3% | 0.77 | 2.03 |
| Trajectron++ with $L_{unlike}$ | 6s | 0.98 | 2.59 | 8.9% | 0.77 | 2.03 |

directly, and our method in Tab.5. Although simply removing the context-violated prediction helps improve the distribution accuracy, our method still performs better.

**Hyperparameter $\gamma$**   In our previous experiments, $\gamma$ is simply set to 1 and turned on smoothly during training. We further performed ablations on $\gamma$ with values including 0.1, 0.3, 3, 10. We observed performance degradation for smaller $\gamma$, and similar performance to $\gamma = 1$ for bigger $\gamma$; see the detailed ablations in the Section 'Hyperparameter $\gamma$' of the Appendix.

**Check-horizon for negative candidates**   Assume we have a 3-second predicted trajectory that obeys our checker but tends to violate it in the near future (e.g., in 1 second). Such a trajectory is able to pass our checker's examination but still unlikely to happen in the real world. To classify it as a negative trajectory, we check the future of this prediction by extending the prediction horizon of the candidate trajectories and checking this extended version. Tab.6 shows ablation studies with extended check-horizon from 3 seconds to 4, 5, 6 seconds to verify whether this mechanism improves our method. Note that the extended horizon is only used for selecting out the negative trajectories. Trajectories are truncated back to 3 seconds when computing the unlikelihood loss $L_{unlike}$. Our method benefits from an adequately extended check-horizon as extending the negative trajectories to 4 seconds improves the FDE Full by about 0.08. Note that when we further extend the check-horizon to 5 seconds and 6 seconds, we do not observe a better performance. This might be because the future of the prediction itself is too inaccurate when we extend the horizon too much and make the judgment of the original prediction unreliable. Numbers are averaged over two training instances.

## 5   Conclusion

We present an unlikelihood guided trajectory prediction method that leverages context information into learning signals by minimizing the probability of unlikely trajectories. During training, our context checker detects predicted unlikely trajectories and their probabilities are minimized using an unlikelihood loss. Our method can be seamlessly incorporated into state-of-the-art distribution-based models with a maximum likelihood estimation objective. Our experimental results demonstrate that our method significantly improves the predicted distribution quality of state-of-the-art trajectory prediction models. We hope that our work may encourage future work on exploring better unlikelihood methods for trajectory prediction and improved context checker models.

# 6    Acknowledgement

This work is funded by a KAUST BAS/1/1685-01-0.

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
