# OpenReview forum: "Motion Forecasting with Unlikelihood Training in Continuous Space"
_robot-learning.org/CoRL/2021/Conference — CoRL2021 Oral_

### Official Review · Reviewer_nXVp · 2021-07-23

**Originality:** Fair
**Technical Quality:** Good
**Clarity Of Presentation:** Very Good
**Impact:** 4

**Recommendation:**

Weak Accept: I recommend accepting the paper, but will not argue for my recommendation if the majority of other reviewers have a different opinion.

**Summary:**

The work introduces the application of 'unlikelihood' loss to augmenting standard training losses for trajectory learning. The work supports the notion of applying (soft) constraint to the training objective by analysing contextual information to inform what should or should not be penalized.

**Issues:**

Comments and Questions
- Connection with MAP: In the abstract the contribution is stated as "Orthogonally, we propose a new objective, unlikelihood training, which forces predicted trajectories that conflict with contextual information to be assigned a lower probability". While the term "unlikelihood" in this context makes perfect sense (and is adopted from prior work - welleck 2019), in my mind what it *really* seems to be in the Bayesian sense is that it is simply a _hyperprior_ in precisely the same sense as in maximum a posteriori (MAP) estimation.  I see that the hyperprior is, in the unliklihood sense, simply another distribution imbued by the "contextual information". Could the authors provide any commentary on a link between the two?

- Unlikelihood data
The authors state "However, these are not given in the dataset. To solve this issue, we approximate the samples by directly drawing a set of trajectories from the predicted distribution and select the trajectories that violate the contextual information out by a context checker." This brings to mind the idea of an adversarial game being played between the model and the checker. Although here the checker is assumed to be a kind of oracle. Would there be any possiblity for encoding uncertainty over the checker itself? Furthermore, it would seem the method is highly dependent on sample complexity. I.e. if the complete set of "invalid" trajectories is sampled uninformatively, then it would seem that the loss would practically contribute not much (since there are no negative samples to learn from). This problem would be exacerbated (potentialy exponentially in higher dimensions). Have the authors considered how such a problem may manifest in the given problems and also how to alleviate such problems (perhaps some kind of informative sampling regime).

- Figure 3. I find the diagram somewhat visually uninformative. This could be improved in many ways: highlighting areas of major differences as given in the caption. At the very least the diagram could be helped with a legend that identifies the various components of interest to leave the reader wondering less (e.g. are there any signficant differences between the blue, green and red distributions? - this is unclear from the photo)

**Reviewer Expertise:**

Good: General knowledge of the area

**Strengths And Weaknesses:**

- The idea is very simple but seems to be effective
- It addresses a seemingly obvious but in my mind absolutely necessary encoding of violations in the application domain
- The paper is well written and the ideas come across clearly

**Summary Of Recommendation:**

Overall I think the paper presents a simple but very practically useful application of the unlikelihood loss for trajectory learning.

---

> ### Author Response · Authors · 2021-08-24
> **Answer to Questions**
>
> We thank Reviewer nXVp for the valuable and helpful comments. We incorporated all the feedback here.
>
> **Connection with MAP: It really seems to be in the Bayesian sense is that it is simply a hyperprior in precisely the same sense as in maximum a posteriori (MAP) estimation. I see that the hyperprior is, in the unlikelihood sense, simply another distribution imbued by the "contextual information". Could the authors provide any commentary on a link between the two?**
>
> Our checker defines a prior distribution implicitly (Let's denote it as $p(Y)$, where $p(Y)=0$ if Y violates the checker and $p(Y)$= positive constant if Y passes), but our method isn't similar to a MAP method.
> When we use the $p(Y)$ as the prior distribution in the MAP framework, the training objective we want to maximize is $p(Y|X) \propto p(X|Y)  p(Y)$. We can observe that we need $p(X|Y)$ here, which we don't have in our model.
> Let's analyse our loss in case we use MLE loss as $L_{orig}$ in Eq.1, setting $\gamma=1$, and assume a ground truth distribution $p_{gt}(Y|X)$ where the real data is from. $L = -E_{p_{gt}(y|X)}\log p_{\theta}(Y|X) + E_{p_{neg}(Y|X)}\log p_{\theta}(Y|X)$.
> Here we have 3 posterior distributions: the real future $p_{gt}(Y|X)$, the posterior negative distribution where the negative trajectories are from $p_{neg}(Y|X)$, and the posterior distribution estimated by the model $p_{\theta}(Y|X)$.
> Due to the expectation over different distribution, the 2 loss terms in $L$ cannot be reduced using Bayes' theorem to get some Bayesian meaning.
> In addition, since the negative trajectories are sampled from the predicted distribution and filtered out by the checker, $p_{neg}(Y|X) = 0$, if $p(Y) \neq 0$; $p_{neg}(Y|X) = p_{\theta}(Y|X)/K$, if $p(Y) = 0$. Here $K$ is a constant to make sure the probability density will be integrated to 1 and $K=\int_{p(Y)=0} p_\theta(Y|X) dY$. Here, the relationship between $p_{neg}(Y|X)$ and $p(Y)$ is not built on Bayesian.
> Therefore, our unlikelhood modeling may not have a strong connection to prior modeling in MAP.
>
>
> **Although here the checker is assumed to be a kind of oracle. Would there be any possibility for encoding uncertainty over the checker itself?**
>
> In future work, we might design a learnable checker if we have the (positive/negative) labels of trajectories. In this case, the checker is trained to do binary classification and the output scores can be viewed as the uncertainty.
>
>
>
> **It would seem the method is highly dependent on sample complexity. I.e. if the complete set of "invalid" trajectories is sampled uninformatively, then it would seem that the loss would practically contribute not much (since there are no negative samples to learn from). This problem would be exacerbated (potentially exponentially in higher dimensions).**
>
>
> Our negative trajectories are sampled from the predicted distribution of the model and filtered by our context checker.
> If there are no so many negative samples, our predicted distribution on this datapoint is already not bad (in the obeying contextual obeying perspective).
> In contrast, if a predicted distribution violates the context a lot, we can easily obtain enough negative trajectories for our method to improve this prediction.
> Since the predicted distribution we use in the experiments is Gaussian, the probability density of the context-violated trajectories will never go to zero.
> We agree that our method might not further reduce the probability density of the negative when it is already too low to easily obtain negative samples.
> A straightforward way to alleviate this issue is to simply sample more. As the number of samples increase, we have a higher chance to obtain negative trajectories with low probability density.
> We might also weigh the negative more during sampling as an informative sample strategy. We leave it for future research.
>
>
> **Figure 3. Highlighting areas of major differences as given in the caption.**
>
> Thank you for your suggestion.
> We added ellipses to highlight 2 context violation cases in Trajectron++ in the updated version.

---

### Official Review · Reviewer_Fbgh · 2021-07-25

**Originality:** Good
**Technical Quality:** Very Good
**Clarity Of Presentation:** Excellent
**Impact:** 4

**Recommendation:**

Strong Accept: I recommend accepting the paper and will argue for my recommendation even if other reviewers hold a different opinion.

**Summary:**

This paper's contrition is to use map information to mitigate false positives in motions forecasting tasks, by minimizing the predictive likelihood of motions that are known to violate known lane directions in a HD map.

**Issues:**

Line 42 "we are the first to propose unlikelihood training for continuous space of trajectories": See R2P2 [A], e.g. fig 6, noting H(q,p) term to penalize bad samples. Might be worth referencing this work and discussing the differences.

Lines 44-46: "inject contextual information as a learning signal" seemed a little vague - could this be clarified?

Grammar line 64: "while Tang and Salakhutdinov [2], Salzmann et al. [1] uses"

Line 63 and 66: small preference against using reference numbers as nouns to begin a sentence.

Line 78-79, could probability delete "in contrast to continuous predictions that represent the future trajectories in our setting" since next sentence says this too.

Line 92: perhaps clarify different between x and X? X is random and x is not?

Line 101: should "Y_{i}" be "y_{i,gt}" (line 92) or "Y_{i,gt}" (line 113) or "\bold{Y}_{i,gt}" (Fig1 caption) or "\bold{y}_{i,gt}" (line 171). Should some of these mean the same thing and should be following consistent notation?

Line 102-103: change "between the predicted distribution and ground truth distribution" --> "between the ground truth distribution and the predicted distribution"

Left quotations on lines 105, 137, 138, 283

I think the connection to REINFORECE in lines 138-142 is tenuous, consider removing.

Line 151: "map-based checker": this is related to the Beelines [B] paper, worth citing and comparing to in the related work, since Beelines also contributes map information to alter the typical ML loss for forecasting in a way that avoids the more obvious-to-humans out-of-bounds errors.

Line 157-158: "The trajectories that fail to pass the test are identified as negative trajectories Y i,neg": I wonder if there's a future work here based on a continuum of being more and more negative, the more the vehicle heading differs from the lane's heading, in addition to binary pass/fail tests.

Equation 4: it's interesting the log_sigma cancelled out on line 174 and now doesn't feature in the sigma gradient term, effectively turning this into an L2 learning task with an initially-set hyperparameter of sigma. I.e. the two Gaussians have in effect lost their "complexity terms" that regularize sigma. I assume one can expect non-equal number of positives and negatives sample in general, so this affect would disappear, unless the positive and negative batches are *averaged* and not *summed*? Possibly worth commenting on this somewhere lines 177-186, if space limitation allow for (not big priority).

Lines 187-194: interesting observation! Nice explanation.

Table 1: what is the "Context-Violation-Rate" of the ground truth trajectories for reference? (presumably something non-zero, but close to zero? Unless the "disable rule" in lines 160-161 ensure it'd be zero exactly?)

Lines 235-237: nice explanation.

Line 284: "and leads to a better prediction" - interesting! Can the standard error be added into Table 5 that shows this, analogous to Table 4 etc?

I like the combination of likely and unlikely trajectories in equation 2, especially since even if there's an error in the context checker, significant likelihood of an other-wise valid motion can override such errors. But this really depends on how is gamma set: how should gamma be set? Edit I see on lines 259 it's scheduled 0 to 1, is there anything special about the number 1 in this context, or could it be anything? (do positive and negative sample matter equally for the purpose of accurate prediction? Could you have got better scores if this was schedule up to a maximum of 0.1 or 10?). Additional experiments not necessarily required, more after an explanation and clarification in the text.

[A]
R2P2: A reparameterized pushforward policy for diverse, precise generative path forecasting
by Nicholas Rhinehart, Kris Kitani, and Paul Vernaza
https://openaccess.thecvf.com/content_ECCV_2018/papers/Nicholas_Rhinehart_R2P2_A_ReparameteRized_ECCV_2018_paper.pdf

[B]
Beelines: Motion Prediction Metrics for Self-Driving Safety and Comfort
by Skanda Shridhar, Yuhang Ma, Tara Stentz, Zhengdi Shen, Galen Clark Haynes, Neil Traft
at ICRA 2021, https://arxiv.org/abs/2011.00393

**Reviewer Expertise:**

Very good: Comprehensive knowledge of the area

**Strengths And Weaknesses:**

Strengths:
 - Motivation
 - Clarity of writing
 - Experiments and multiple real datasets used

Weaknesses:
 - Context checker is not learned, and some of the closest related works were not cited. While the learned part of this work is the original predictive model, and the context checker provides a superior loss function, as the authors demonstrate, the rules are handcrafted (e.g. binary 90 degree rule on lines 156-157), which makes this difficult to immediately extend to anything else, like cyclists, pedestrians, etc. While the point of this paper is not necessarily this, but to open research up to unlikelihood training in motion forecasting, other methods have used learning based approaches to mitigate false positives for autonomous vehicle motion in ways that are less hand-crafted and more widely applicable (see R2P2 algorithm, cited below [A]). In addition, the Beelines paper (cited below [B]) is very similar by using map information to alter the loss function for motion forecasting, and should be discussed in the context of this work.

**Summary Of Recommendation:**

I recommend weak-accept in light of the "strengths and weaknesses" discussion above. However, willing to consider strong-accept based on author responses. This is an important area that the community should be looking into more, and comparing against related works would help strengthen this paper.

---

> ### Author Response · Authors · 2021-08-24
> **Answer to Questions Part 1**
>
> Thank you for your detailed and helpful review, questions and suggestions. We here address them and incorporated all the feedback.
>
>
> **Context checker is not learned**
>
> Training a context checker to judge whether a trajectory is 'positive' or 'negative' needs labels. (E.g., Ziegler et al. [https://arxiv.org/pdf/1909.08593.pdf], but in the field NLP)
> However, current datasets don't annotate and provide negative trajectories.
> A straightforward way to bypass this issue is labeling negative trajectories based on some rules and train the checker on it.
> However, we think in this case the learned checker is trained to approximate the rules.
> Therefore, treating directly these rules as the checker is a better way. Besides, we don't introduce any additional learnable parameters and keep the training simple.
> We think that a learned checker might be more 'intelligent' than our handcrafted checker if we have proper human-annotated data. We leave this for future work.
>
> **The rules are handcrafted (e.g. binary 90-degree rule on lines 156-157), which makes this difficult to immediately extend to anything else, like cyclists, pedestrians, etc.**
>
> Indeed, our current rules are designed for vehicles.
> However, the overall framework of our method is not limited to the rules we proposed.
> For example, we might check whether the predicted trajectories of a pedestrian are on the sidewalk or in front of a moving vehicle. There is a room for future work to improve the checker to better extend to cyclists and pedestrians.
>
>
>
> **Other methods have used learning-based approaches to mitigate false positives for autonomous vehicle motion in ways that are less hand-crafted and more widely applicable. R2P2, e.g. fig 6, noting H(q, p) term to penalize bad samples. Might be worth referencing this work and discussing the differences.**
>
> Thank you for raising up this point!
> Our method is quite different from R2P2.
> The q term in R2P2 is the predicted distribution and the p term is the ground truth distribution.
> The H(q, p)  loss term  contains the prediction and the ground truth future only and hence
> it is still based on the similarity between the prediction and the ground truth.
> Indeed, the ground truth distribution $p(y|x)$ should fit the context quite well.
> However, we don't have it.
> R2P2 train another distribution $\tilde p(y|x)$ using the ground truth trajectory as the surrogate of $p(y|x)$.
> And we will need to formulate how to train the surrogate well so the surrogate obeys the context.
> As shown in the Eq.9 of R2P2, the training objective of the surrogate is another MLE loss based on the ground truth only. Therefore, this surrogate itself might have the same MLE issue we mention in the paragraph 'Limitation of MLE on Motion Forecasting' of Section 3.1.
> In contrast, our unlikelihood loss is constructed from the context checker. Therefore, it is based on the context information directly (like the road boundary and the driving direction) instead of only ground truth information.
> We updated our related work section to include R2P2 and a brief comparison.
>
>
>
>
> **The Beelines paper is very similar by using map information to alter the loss function for motion forecasting and should be discussed in the context of this work.**
>
> We notice that the Beelines paper focuses on the evaluation only and doesn't propose a loss for training.
> This paper proposes 2 metrics to quantify the effects of motion prediction on safety and comfort. These metrics are based on estimating potential collision between the ego vehicle and the surrounding vehicle.
> This paper shows that these 2 metrics are better than L2-error-based metrics like ADE and FDE in identifying unsafe events.
> Since we focus on training a model instead of evaluation, we think our method is not very similar to Beelines.
> However, we still think Beelines is interesting.
> Although it's unclear whether the Beelines metrics can be used as a loss during training,
> it's possible to integrate these metrics into our framework as an additional checker rule.
> Our current checker rules focus on road boundaries and lane direction, but not the potential future collision.
> These metrics can complement our rules.
> Note that the Beelines metrics are complicated but the code is not opensourced.
> Hence we could not reliably use their proposed metrics to evaluate our method.
> We added a short discussion in the Context Checker Design paragraph to show the integration possibility in the updated version.

---

> > ### Comment · Reviewer_Fbgh · 2021-08-30
> > **Reviewer Reply**
> >
> > ### Context checker not learned / R2P2:
> >
> > I partly disagree that negative examples are required for learned context checkers. It's true that R2P2 trains an surrogate p(y|x) using logistic regression in eq 9, although the regression is trained to be spatial only (to define a map-cost), independent enough from the original task for mitigate false positives (see Fig 7 comparison). Inverse reinforcement learning methods would be another way to learn context checker (example: maximum causal entropy IRL, summarized by eq 1 in GAIL https://arxiv.org/pdf/1606.03476.pdf).
> >
> > Nevertheless, I appropriate the author's responses and changes to the text to compare with related literature work, and have updated my score.
> >
> > ### Other
> >
> > I've read the rest of the (3 parts) response, and thank the authors for the additional clarify.

---

> ### Author Response · Authors · 2021-08-24
> **Answer to Questions Part 2**
>
> **Lines 44-46: "inject contextual information as a learning signal" seemed a little vague - could this be clarified?**
>
> The original MLE loss is built using the ground truth trajectory only.
> Therefore, the model is trained to be close to the ground truth only. Although leveraging the contextual information helps to achieve this goal in some cases, the training objective doesn't directly force the model to take the contextual information into account.
> There is no contextual signal in the MLE loss.
> In contrast, our unlikelihood training loss is based on the context checker that depends on the contextual information. Therefore, our training objective contains contextual information. We have updated Line 44-46 (blue part in the updated paper) to make it more clear.
>
>
>
>
> **Line 92: perhaps clarify different between x and X? X is random and x is not?**
> **Line 101: should "${Y_{i}}$" be "$y_{i,gt}$" (line 92) or "$Y_{i,gt}$" (line 113) or "$\textbf{Y}{i,gt}$" (Fig1 caption) or "$\textbf{y}{i,gt}$" (line 171). Should some of these mean the same thing and should be following consistent notation?**
>
> Sorry for the confusion of the notation in Section 3.1. All of the above-mentioned $Y$ should be $Y_{i, gt}$. And $x$ in Line 92 should be $X$.
> We revised Section 3.1 and now they have the correct notation.
>
>
> **Line 157-158: "The trajectories that fail to pass the test are identified as negative trajectories Y i,neg": I wonder if there's a future work here based on a continuum of being more and more negative, the more the vehicle heading differs from the lane's heading, in addition to binary pass/fail tests.**
>
> This is an interesting direction. A learnable checker might be close to this idea since it does binary classification and the output continuous score can be used as a 'negative level'.
> In our future work, we might consider using such a negative level as a continuous reward function to train the model.
>
>
>
>
> **Equation 4: it's interesting the log sigma canceled out on line 174 and now doesn't feature in the sigma gradient term, effectively turning this into an L2 learning task with an initially-set hyperparameter of sigma. I.e. the two Gaussians have in effect lost their "complexity terms" that regularize sigma. I assume one can expect non-equal numbers of positives and negatives sample in general, so this effect would disappear unless the positive and negative batches are averaged and not summed? Possibly worth commenting on this somewhere lines 177-186, if space limitation allows for (not a big priority).**
>
> We agree with the observation but our model still regularizes $\sigma$ differently.
> Let's first check the gradient of the MLP loss w.r.t. $\sigma$. $\frac{d L}{d \sigma} = \frac{1}{\sigma} - \frac{1}{\sigma^3}  (y - \mu)^2$   As you can see the first term (which is from the derivate of log sigma square term) is always positive and the second term is always   negative or zero (note that $\sigma > 0$). Therefore, $\log \sigma^2$ serves as a regularizer of the $\sigma$.
> However, although $\log \sigma^2$ is canceled out in our case, our loss gradient (Eq.4) is not always zero or negative. The sign of the gradient actually depends on the distance between the predicted center and the ground truth (let's denote it as $d_{pos}$), and the distance between the predicted center and the negative (denoted as $d_{neg}$). These 2 distances together serve as regulators to control how $\sigma$ changes in this case.
> When $d_{pos} > d_{neg}$, our prediction is far away from the ground truth, so $\sigma$ increases to cover it.
> When $d_{pos} < d_{neg}$, $\sigma$ decreases to exclude the context-violation region from high probability density region of the distribution.
> During training, our positive and negative batches are averaged and not summed (As Eq.1 is an expectation over the negatives).
>
>
>
>
>
>
> **Table 1: what is the "Context-Violation-Rate" of the ground truth trajectories for reference? (presumably something non-zero, but close to zero? Unless the "disable rule" in lines 160-161 ensures it'd be zero exactly?)**
>
> We updated Table 1 and 3 to include the violation rate of the ground truth. In nuScenes, it is 5.44\%. Argoverse is 0.85\%.
>
>
>
> **Line 284: "and leads to a better prediction" - interesting! Can the standard error be added into Table 5 that shows this, analogous to Table 4 etc?**
>
> Thank you for this suggestion. We have integrated the standard error into Table 5

---

> ### Author Response · Authors · 2021-08-24
> **Answer to Questions Part 3**
>
> **How should gamma be set? I see on Lines 259 it's scheduled 0 to 1, is there anything special about the number 1 in this context, or could it be anything?**
>
> In our main paper, we simply set it to 1 and noticed a good performance. We add new ablation experiments with different gamma values (0.1, 0.3, 3, 10) in the section 'Hyperparameter gamma' of the supplementary material and briefly introduce it in the ablation study of the main paper.
> We notice that if we use a small $\gamma$, the improvement over the original Trajectron++ is decreased.
> This is expected since a smaller $\gamma$ reduces the effect of our unlikelihood loss.
> The performance doesn't change much when we further increase $\gamma$ from 1 to 3 and 10.
> However, we notice that when we set $\gamma$ to 3 and 10, as the training goes, the prediction accuracy decreases after it reaches the best performance.
> We think this is because the model focuses too much on obeying the context and pays less attention to getting close to the ground truth in these cases.
> In conclusion, balancing the original training loss and our likelihood loss matters, and simply setting $\gamma$ to 1 could balance the original training loss and our unlikelihood loss well.
>
>
>
> **Grammar line 64: "while Tang and Salakhutdinov [2], Salzmann et al. [1] uses"**
>
> **Line 63 and 66: small preference against using reference numbers as nouns to begin a sentence.**
>
> **Line 78-79, could probability delete "in contrast to continuous predictions that represent the future trajectories in our setting" since the next sentence says this too.**
>
> **Line 102-103: change "between the predicted distribution and ground truth distribution" --> "between the ground truth distribution and the predicted distribution"**
>
> **Left quotations on lines 105, 137, 138, 283**
>
> **I think the connection to REINFORECE in lines 138-142 is tenuous, consider removing.**
>
> Thank you for these valuable suggestions on writing! We have updated our paper to include all the suggestions.

---

### Official Review · Reviewer_XGcM · 2021-07-26

**Originality:** Very Good
**Technical Quality:** Very Good
**Clarity Of Presentation:** Excellent
**Impact:** 4

**Recommendation:**

Strong Accept: I recommend accepting the paper and will argue for my recommendation even if other reviewers hold a different opinion.

**Summary:**

One of the problems in trajectory prediction is that the predicted trajectories do not abide by its context, i.e., they can go out of drivable areas. To solve this issue, the authors proposed an additional "unlikelihood" loss function which discourages the likelihood term to put lower density near negative trajectories. The authors show the improvement on nuScenes dataset where the loss is added to Trajectron++, and Argoverse dataset where the loss is added to Gaussian LaneGCN. The ablation suggests that this approach performs better than an adhoc post-processing approach that removes violation.

**Issues:**

I'd like to see the authors address the 3 points mentioned above in the weakness section.
Some word choices: geometrical nearness --> geometrical proximity

**Reviewer Expertise:**

Good: General knowledge of the area

**Strengths And Weaknesses:**

Strengths
1. I like the key idea is based on the fact that validating trajectories is easier than generating them. So, we can first classify good and bad trajectories and use those to shape the output distribution.
2. The idea of using "unlikelihood" seems to be novel, and this paper can inspire future avenues in exploiting negative trajectories.
3. The ablation and the results are convincing.

Weaknesses
1. The loss function used in the experiment is not clear to me. For example in eq 5, what does p_theta(Y|X,z) looks like? Is it a gaussian with learned mean and variant?
2. How does metrics reflect the multimodal nature of the trajectories? L2 measurement alone does not capture how accurate we model Y|X when they are multimodal.
3. Figure 1 is not very clear in demonstrating the issue the paper is trying address. The uncertainty is only slightly off the track. Also, it seems like the high uncertainty is more of a problem in this figure. Also it would be nice to compare the result of this method before and after in in figure 1.
4. How sensitive is the result to the hyper-parameter? Is the result overfit to the validation data -- as the paper mentioned that the validation data are used for testing?

Others:
1. There might be similar approaches in ML+motion planning to avoid constraint violation.


**Summary Of Recommendation:**

The paper provides a simple trick that can be added generative models for motion forecasting. The paper provides convincing results that it can improve over the state-of-the-arts. However, I still have slight concern over the hyper-parameter tuning and test settings.

---

> ### Author Response · Authors · 2021-08-24
> **Answer to Questions**
>
> Thank you for your detailed and helpful review, questions and suggestions. We here address them and incorporated all the feedback.
>
> **Loss function used in the experiment is not clear to me. For example in eq 5, what does $p_\theta(Y|X,z)$ looks like? Is it a gaussian with a learned mean and variant?**
>
> $p_\theta(Y|X)$ depends on the type of the distribution the model we combined with uses.
> In case of a Gaussian mixture model (which is used by Trajectron++ as mentioned in ),
> $\log p_\theta(Y|X)= \sum_t \log \sum_z p(z|X) N(y_t|\mu_{z, t}, \Sigma_{z, t})$. Here, $\mu_{z,t}$ and $\Sigma_{z,t}$ are the predicted mean and the covariance at the step t of the z-th component. $p(z|X)$ is a predicted categorical distribution.
> Gaussian LaneGCN generates 6 Gaussian distributions. During training, Gaussian LaneGCN selects the best distribution among these 6 for each datapoint and optimizes the MLE loss (and the unlikelihood loss in our case) on it.
> So $p_\theta(Y|X)$ here is simply a Gaussian with a learned mean and variance.
>
>
> **Figure 1 is not very clear in demonstrating the issue the paper is trying to address. The uncertainty is only slightly off the track. Also, it seems like the high uncertainty is more of a problem in this figure. Also, it would be nice to compare the result of this method before and after in figure 1.**
>
> Thanks for this suggestion! To better demonstrate, we replace the original Fig.1 with a crop version of the Trajectron++ example in Fig.3 and highlight the violation part by ellipses; see Fig.3. More qualitative examples are listed in the supplementary materials.
>
>
>
>
>
> **How do metrics reflect the multimodal nature of the trajectories? L2 measurement alone does not capture how accurate we model Y|X when they are multimodal.**
> We target improving the quality of the predicted distribution. However, we only have 1 ground truth future trajectory instead of the real future distribution.
> In this case, it is hard to have a metric that estimates how similar our predicted distribution $p_\theta(y|x)$ is to the ground truth distribution $p_{gt}(y|x)$.
> As our unlikelihood loss is built upon the context checker and is not so related to the multimodal property, we assume the multimodal nature of the prediction when using our loss will not be obviously affected.
> In this case, the reduced L2 measurement like FDE-Full and ADE-Full can still show the improvement of the distribution quality.
> In addition, we also report the context-violation rate in our experiments.
> This metric is compatible with the multi-modal nature and shows that our distribution quality is improved.
>
>
>
>
> **How sensitive is the result to the hyper-parameter? Is the result overfit to the validation data -- as the paper mentioned that the validation data are used for testing?**
>
> For nuScene dataset, we notice that the original hyperparameters that Trajectron++ uses are not good enough, and we are afraid that this will lead to a 'fake improvement' of our method. Therefore, we tune the hyperparameters of Trajectron++ based on  **the performance of Trajectron++** (as mentioned in Line 215-216 of the updated paper) on a validation set that is split from the original training set. The original validation set is used as the test set, and no hyperparameters are optimized on it (i.e.,  it is disjoint from the hyper-parameter validation set ). Our method has a new hyperparameter gamma compared to Trajectron++. We simply set it to 1 in our experiments.
> For Argoverse, the hyperparameters of Gaussian LaneGCN and our method are the same as the default hyperparameter of LaneGCN. For our additional gamma, we similarly set it to 1.
> We did run new experiments with different gamma values (0.1, 0.3, 3, 10) and added the detailed results and discussion in the appendix (Section 'hyperparameter gamma'). We also briefly introduce this experiment in the ablation study of the main paper.
> We notice that if we use a small $\gamma$, the improvement over the original Trajectron++ is decreased.
> This is expected since a smaller $\gamma$ reduces the effect of our unlikelihood loss.
> The performance doesn't change much when we further increase $\gamma$ from 1 to 3 and 10.
> However, we notice that when we set $\gamma$ to 3 and 10, as the training goes, the prediction accuracy decreases after it reaches the best performance.
> We think this is because the model focuses too much on obeying the context and pays less attention to getting close to the ground truth in these cases.
> In conclusion, balancing the original training loss and our likelihood loss matters, and simply setting $\gamma$ to 1 could balance the original training loss and our unlikelihood loss well.

---

> > ### Comment · Reviewer_XGcM · 2021-09-01
> > **Thank you**
> >
> > Thank you for addressing my questions and concerns. After reading the author's response, I'd like to maintain my score which is strong accept.

---

### Author Response · Authors · 2021-08-24
**Paper Revised Version**

We thank reviewers for their valuable and insightful feedback. We are encouraged that they find our idea is novel (XGcM), interesting (Fbgh), and simple but effective (nXVp); our writing is clear (XGcM, Fbgh, nXVp), and our work could be impactful (XGcM, Fbgh, nXVp). We here reply to the reviewers' comments individually and incorporate all the feedback.

**Updated Paper**
We also revised our paper and supplementary material to incorporate reviewers' feedback. The changed text is marked in blue color for an easy and quick check. We list here the key updates that we applied to improve the paper.

1. We added new ablation studies on our hyperparameter $\gamma$ in both the main paper and the supplementary material.

2. We added a new discussion with the R2P2 method in the related work section and Beelines in the Context-Checker Design paragraph in Sec.3.3

3. We updated Fig.1 and Fig.3 for a better demonstration

4. We added the context-violation rate of the ground truth in Tab.1 and Tab.3.

5. We added the standard deviation in Tab.5.

6. We fixed grammar and notation mistakes.

---

### Public Comment · ~Yuejiang_Liu1 · 2021-09-15
**Similar recent work**

Glad to read your paper in the context of vehicle motion forecasting.

In fact, we reported a similar idea, entitled [Social-NCE](https://arxiv.org/abs/2012.11717), on Arxiv at *the end of last year* (later accepted to ICCV this year).

While our work was more focused on human motion, the key problem we aimed to address (i.e. reducing implausible / unlikely trajectory outputs) and the solution we proposed (i.e. adding an auxiliary loss to incorporate augmented negative examples) share quite significant similarities with what's presented in your paper.

The motivation noted in your Line 128 `Inspired by the positive and negative data pairs in contrastive learning and unlikelihood training` also indicates this connection. As far as I understand, the key difference between our papers lies in the form of auxiliary loss - we picked the former `contrastive loss`, whereas yours adopts the latter `unlikelihood loss`. Nevertheless, they are essentially not much different, as analyzed in earlier work (e.g. [word2vec Explained](https://arxiv.org/pdf/1402.3722.pdf)) as well as recent literature in contrastive learning.

We'd be glad to see your thoughts on it.

---

> ### Author Response · Authors · 2021-09-16
> **Reply**
>
> Thank you for your interest in our work!
> An [initial version](https://openreview.net/forum?id=4JLiaohIk9) of our work is publicly accessible since the end of Sept 2020 and it is 2 months earlier than the release of Social-NCE.
> While both methods try to reduce unlikely trajectories by introducing a new loss based on the negative (Actually, [R2P2](https://openaccess.thecvf.com/content_ECCV_2018/papers/Nicholas_Rhinehart_R2P2_A_ReparameteRized_ECCV_2018_paper.pdf) (ECCV2018, earlier than both of our works) can also be interpreted into this class), there are some big differences between these 2 methods.
>
> 1. Social-NCE regularizes the learned *representations* (as contrastive learning targets on better representation). Instead, our method improves the predicted *distribution* quality (as unlikelihood learning target on better distribution) by directly reducing the likelihood of trajectories sampled from a negative distribution. Therefore, our method doesn't add any additional learnable parameters like the event encoder in Social-NCE to make it work.
> 2. We build the negative distribution based on both our context-checker (this is based on context information like the driving direction) and *our prediction*. This guarantees that the negative samples come from the model itself, and the model learns from the mistake it made. Social-NCE views the neighborhood as the negative sample.
>
> It is a joy to read your paper in the context of pedestrian motion prediction. We are happy to see that more and more works focus on alleviating the negative prediction issue.

---

> > ### Public Comment · ~Yuejiang_Liu1 · 2021-09-16
> > **Thanks for response**
> >
> > Thanks for your detailed response! We were not aware of your initial version on OpenReview until now. Indeed, our work was to regularize the learned representation, as opposed to the estimate of trajectory likelihood considered in your paper. Again, it's great to see our concurrent findings about the advantage of modeling negative examples in motion problems.

---

### Meta-Review · Area_Chair_i3gA · 2021-08-14

**Recommendation:** Accept (Oral)
**Confidence:** 4

**Metareview:**

Summary:
This paper presents a method to improve the quality of agent predictions by incorporating a contextual loss term that motivates unrealistic predictions (e.g. violate lane direction) to be assigned lower likelihoods. The method is validated on two benchmark scene sets.

Clarity: All reviewers commend the paper for its clear presentation. The paper flowed well and presented a well-motivated, well-defined piece of technical work.

Significance: Reviewers strongly agree that this paper pushes an area of work that could be extremely impactful to the community.

Quality: The experimental results are overall convincing. In fact, reviewers nXVp and XGcM both note that the proposed method seems simple, however, provides convincing improvements to algorithm performance. This gives the method a high likelihood of being adopted by the research community.

Originality: There was mixed feedback about the novelty of the proposed approach, and the paper could improve in articulating its contribution. (See “Opportunities for improvement” for more details).

Pros:
This paper presents a method that is philosophically compelling in principle and whose utility is backed up by comprehensive simulation experiments. The paper is also very well-written; readers are able to grasp the main ideas right away and see those main ideas come through in the experimental section.

Main opportunities for improvement:
There are two main areas that can be improved to further solidify this paper.

The first is orienting the proposed work better in current literature. Reviewers mention a number of related papers and methods; this paper should explicitly describes similarities and differences between itself and these existing techniques. Reviewers Fbgh and nXVo provide specific examples of missing related works.

The second is to refine the experimental results section. The revision should expand on 1. implementation details the should be clarified and 2. further analyses and explanations of the existing results. All reviewers, and in particular reviewer Fbgh, make specific suggestions for what needs to be addressed.

Thank you for considering our feedback, and we look forward to seeing the updated paper.

================ Final Decision

All reviewers recommended accept. I decided to recommend oral presentation because after revisions, reviewer Fbgh updated their score to strong accept as well, the paper has strong scores for impact (4s across the board), extremely strong scores for technical quality and clarity across the board, and positive originality scores. In general, the paper seems to be very well received.

---

### Decision · Program_Chairs · 2021-09-13

**Decision:**

Accept (Oral)

**Comment:**

Summary:
This paper presents a method to improve the quality of agent predictions by incorporating a contextual loss term that motivates unrealistic predictions (e.g. violate lane direction) to be assigned lower likelihoods. The method is validated on two benchmark scene sets.

Clarity: All reviewers commend the paper for its clear presentation. The paper flowed well and presented a well-motivated, well-defined piece of technical work.

Significance: Reviewers strongly agree that this paper pushes an area of work that could be extremely impactful to the community.

Quality: The experimental results are overall convincing. In fact, reviewers nXVp and XGcM both note that the proposed method seems simple, however, provides convincing improvements to algorithm performance. This gives the method a high likelihood of being adopted by the research community.

Originality: There was mixed feedback about the novelty of the proposed approach, and the paper could improve in articulating its contribution. (See “Opportunities for improvement” for more details).

Pros:
This paper presents a method that is philosophically compelling in principle and whose utility is backed up by comprehensive simulation experiments. The paper is also very well-written; readers are able to grasp the main ideas right away and see those main ideas come through in the experimental section.

Main opportunities for improvement:
There are two main areas that can be improved to further solidify this paper.

The first is orienting the proposed work better in current literature. Reviewers mention a number of related papers and methods; this paper should explicitly describes similarities and differences between itself and these existing techniques. Reviewers Fbgh and nXVo provide specific examples of missing related works.

The second is to refine the experimental results section. The revision should expand on 1. implementation details the should be clarified and 2. further analyses and explanations of the existing results. All reviewers, and in particular reviewer Fbgh, make specific suggestions for what needs to be addressed.

Thank you for considering our feedback, and we look forward to seeing the updated paper.

================ Final Decision

All reviewers recommended accept. I decided to recommend oral presentation because after revisions, reviewer Fbgh updated their score to strong accept as well, the paper has strong scores for impact (4s across the board), extremely strong scores for technical quality and clarity across the board, and positive originality scores. In general, the paper seems to be very well received.